# Immediate Hypersensitivity Reactions Induced by COVID-19 Vaccines: Current Trends, Potential Mechanisms and Prevention Strategies

**DOI:** 10.3390/biomedicines10061260

**Published:** 2022-05-28

**Authors:** Shuen-Iu Hung, Ivan Arni C. Preclaro, Wen-Hung Chung, Chuang-Wei Wang

**Affiliations:** 1Cancer Vaccine & Immune Cell Therapy Core Laboratory, Department of Medical Research, Chang Gung Memorial Hospital, Linkou Branch, Taoyuan 333, Taiwan; hungshueniu@gmail.com; 2Institute of Pharmacology, School of Medicine, National Yang Ming Chiao Tung University, Taipei 112, Taiwan; 3Drug Hypersensitivity Clinical and Research Center, Department of Dermatology, Chang Gung Memorial Hospital, Linkou 333, Taiwan; ivanpreclaro@gmail.com; 4Drug Hypersensitivity Clinical and Research Center, Department of Dermatology, Chang Gung Memorial Hospital, Taipei 105, Taiwan; 5Chang Gung Immunology Consortium, Chang Gung Memorial Hospital, Chang Gung University, Taoyuan 333, Taiwan; 6Department of Dermatology, Xiamen Chang Gung Hospital, Xiamen 102218, China; 7Whole-Genome Research Core Laboratory of Human Diseases, Chang Gung Memorial Hospital, Keelung 204, Taiwan; 8School of Clinical Medicine, Tsinghua University, Beijing 100190, China; 9Department of Dermatology, Ruijin Hospital, School of Medicine, Shanghai Jiao Tong University, Shanghai 200240, China; 10Genomic Medicine Core Laboratory, Chang Gung Memorial Hospital, Linkou 333, Taiwan

**Keywords:** COVID-19 vaccines, IgE-mediated pathway, immediate hypersensitivity reactions, skin test

## Abstract

As the world deals with the COVID-19 pandemic, vaccination remains vital to successfully end this crisis. However, COVID-19-vaccine-induced immediate hypersensitivity reactions presenting with potentially life-threatening systemic anaphylactic reactions are one of the reasons for vaccine hesitancy. Recent studies have suggested that different mechanisms, including IgE-mediated and non-IgE-mediated mast cell activation, may be involved in immediate hypersensitivity. The main culprits triggering hypersensitivity reactions have been suggested to be the excipients of vaccines, including polyethylene glycol and polysorbate 80. Patients with a history of allergic reactions to drugs, foods, or other vaccines may have an increased risk of hypersensitivity reactions to COVID-19 vaccines. Various strategies have been suggested to prevent hypersensitivity reactions, including performing skin tests or in vitro tests before vaccination, administering different vaccines for the primary and following boosters, changing the fractionated doses, or pretreating the anti-IgE antibody. This review discusses the current trends, potential mechanisms, and prevention strategies for COVID-19-vaccine-induced immediate hypersensitivity reactions.

## 1. Introduction

Since its discovery, coronavirus disease 2019 (COVID-19) has remained a global public health pandemic [1]. With the announcement of its genetic sequence, researchers and companies have raced to develop vaccines to end the pandemic. The administration of vaccines has successfully reduced the morbidity and mortality of COVID-19 [2,3,4,5]. According to the World Health Organization [6], the increasing availability and utilization of vaccines effectively protects people from the disease severity [7,8,9]. Currently, four classes of vaccines against COVID-19 are available. (1) mRNA vaccines use an innovative approach for inducing messenger RNA (mRNA) molecules to safely produce COVID-19 proteins, resulting in an immune response. (2) Viral vector vaccines use genetically engineered viral vectors to produce COVID-19 proteins to stimulate the host’s immunity. (3) Inactivated virus vaccines use a weakened state of the COVID-19 virus that the host is capable of mounting an immune response against. (4) Protein subunit vaccines use COVID-19 protein fragments as a stimulus to trigger immune responses [10].

## 2. Clinical Phenotypes of Vaccine-Induced Immediate Hypersensitivity Reactions

Although vaccination has dramatically improved the control of COVID-19 transmission [11], vaccination hesitancy remains a significant issue owing to adverse reactions, particularly unpredictable hypersensitivity reactions [12,13]. Most hypersensitivity reactions to vaccines occur immediately and abruptly within minutes to hours after administration [14,15,16]. The clinical manifestations may range from mild cutaneous eruptions, such as urticaria or angioedema, to life-threatening systemic anaphylaxis [17]. Urticaria is characterized by transient wheal formation and may produce an itching or burning sensation. Angioedema is characterized by painful swelling in the deep dermis and subcutis layers of the skin. Both presentations are part of a spectrum of systemic symptoms, including anaphylaxis [18]. Anaphylaxis is rare but frequently leads to death [19,20].

Most immediate hypersensitivity reactions have occurred after administrating the first dose. However, reactions after the second dose of the COVID-19 vaccine have also been reported [21]. Approximately 86% of anaphylaxis cases induced by COVID-19 vaccines occur within 30 min of inoculation. On the contrary, the onset of other symptoms, such as urticaria, often happens within 3–8 days of the first dose and 2–5 days after the second dose [21,22,23].

Many vaccine-induced hypersensitivity reactions could not be confirmed and have been attributed post factum to alternative diagnoses, such as vasovagal syncope, vocal cord dysfunction, exacerbation of existing chronic spontaneous urticaria, and anxiety. Using an updated global standard for case definitions and guidelines for hypersensitivity reactions following vaccinations may help with clinical differential diagnosis and management [24,25].

## 3. Epidemiology of Immediate Hypersensitivity Induced by Vaccines

Vaccine-induced anaphylaxis cases are estimated to occur in approximately 1 case per 15 million to 2 cases per million individuals [14]. Micheletti F. et al. reported that the risk of anaphylaxis after vaccination in children and adults was estimated to be 1.31 (95% confidence interval [CI], 0.90~1.84) per million doses before the COVID-19 pandemic [26]. The authors identified 33 confirmed vaccine-triggered anaphylaxis cases in the study after 25,173,965 vaccine doses [26]. Among the patients with vaccine-induced immediate hypersensitivity reactions, approximately 66% had urticaria, and 10% had angioedema [27].

For COVID-19 vaccines, cutaneous reactions were reported by 1.9% of individuals after receiving the first dose of an mRNA COVID-19 vaccine. Approximately 2.3% of those who had no adverse events following the first dose developed hypersensitivity reactions after receiving the second dose [28]. Based on a U.S. study, cutaneous reactions induced by the mRNA COVID-19 vaccines were more common in women than in men (85% vs. 15%, *p* < 0.001) [28]. Furthermore, the estimated incidence rates for anaphylaxis in the U.S. were 11.1 cases per million doses administered with the BNT162b2 (Pfizer-BioNTech) vaccine and 2.5 cases per million doses administered with the mRNA-1273 (Moderna) vaccine [16,29,30,31]. The vaccine adverse event reporting system (VAERS) [32] showed that there were 1592 urticaria cases among 15703 (10.13%) cases with adverse reactions, 32 (4.92%) out of 650 adverse event cases of angioedema, and 66 (3.54%) out of 1867 adverse event cases of anaphylaxis from 2020 to January 2022 attributed to COVID-19 vaccines.

A recent meta-analysis study suggested that the estimated incidence of COVID-19-vaccine-induced anaphylaxis ranged from 2.5 to 7067 per one million individuals receiving mRNA COVID-19 vaccines, with an overall pooled prevalence estimate of 5.58 (95% CI, 3.04–8.12; I^2^  =  76.32%, *p* < 0.01) [21]. In contrast, the incidences of nonanaphylactic reactions to mRNA COVID-19 vaccines ranged from 10.6 to 472,973 per one million, with an overall pooled prevalence estimate of 89.53 (95% CI, 11.87–190.94; I^2 ^ =  97.08%, *p* < 0.01) [21]. Chu, DK. et al. performed a meta-analysis of 22 studies, including 1366 patients, and found a low incidence (0.16%) of immediate severe allergic reactions associated with the second dose of the mRNA COVID-19 vaccine among individuals who had an allergic history of their first dose [33]. In a separate study, the incidence rates of anaphylaxis were lower for the viral COVID-19 vaccine (odds ratio [OR], 0.47; 95% CI, 0.33–0.68) and the inactivated COVID-19 (OR, 0.31; 95% CI, 0.18–0.53) vaccine [34]. Different setups of studies may observe different incidence rates. Table 1 lists the incidence rates of anaphylactic and nonanaphylactic hypersensitivity reactions to COVID-19 vaccines.

The available information suggests that the incident rate of adverse events after the administration of the protein-based vaccine (Nuvaxovid/NVX-CoV2373 produced by Novavax, Gaithersburg, MD, USA) is lower than the mRNA vaccines [43,44,45]. Almost all the reported incidences of vaccine-induced adverse reactions come from passive reporting systems (such as VAERS), which may underestimate the true burden [46]. In addition, limited prospective studies have been performed, which could result in a much higher rate of acute allergic reactions, possibly due to a nocebo effect [47].

## 4. Causality of Vaccine-Induced Immediate Hypersensitivity Reactions

Vaccine excipients and active components could cause allergens to elicit hypersensitivity reactions. These antigen components, such as toxoids or constituents of pneumococcal vaccines, cause symptoms ranging from urticaria to anaphylaxis. Hypersensitivity reactions may be induced when patients receive the first or the second dose of a vaccine [48,49].

Vaccine excipients are known to be ingredients other than the active components of vaccines. These are inactive ingredients that stabilize or preserve the viability of the vaccines and maintain their bioavailability. Egg and ovalbumin (a residual component of egg processing) are considered the most frequent food allergies in children and the most suspected culprits for allergies induced by the administration of traditional vaccines [49,50,51,52]. Gelatin is another culprit excipient for vaccine-induced immediate hypersensitivity reactions [51,52,53].

Vaccine adjuvants are also possible allergens [54]. Aluminum hydroxide and aluminum phosphate are adjuvants that are more commonly found in vaccines but are not in the COVID-19 vaccine. Although rare, they are commonly associated with delayed-type hypersensitivity reactions. Aluminum can also induce immediate-type hypersensitivity by stimulating mast cells and other immune cells [49,55]. Another vaccine adjuvant, AS03, is a squalene derivative that is incorporated into influenza vaccines. Epidemiological studies in Canada have shown an approximately 20-fold increase in the incidence of immediate hypersensitivity using AS03-adjuvanted vaccines compared with non-AS03 vaccines. The immune mechanism underlying vaccine-adjuvant-induced immediate hypersensitivity reactions remains unclear [49,54,56,57].

## 5. Proposed Immune Mechanisms for Vaccine-Induced Immediate Hypersensitivity Reactions

According to cellular and molecular features defined by Gell and Coomb, there are four types of hypersensitivity reactions: I, II, III, and IV [58]. Type I hypersensitivity reactions involve IgE-mediated immune responses and occur rapidly after exposure to allergens. Type II hypersensitivity is mediated by IgG or IgM antibodies, and type III hypersensitivity involves the immune complexes. Type IV hypersensitivity is mediated by T lymphocytes, also known as delayed-type reactions.

Mast cells are considered the most critical immune cells responsible for immediate hypersensitivity reactions, as they secrete various inflammatory cytokines and induce various systemic immune responses [52]. There are four proposed mechanisms for immediate hypersensitivity reactions, including (1) immunoglobulin E (IgE)-mediated, (2) complement-receptor-mediated, (3) MRGPRX2 (Mas-related G-protein coupled receptor member X2)-mediated mast cell direct activation, and (4) an unknown mechanism (Figure 1).

The IgE-dependent pathway is the most common and well-known mechanism [52]. In IgE-mediated hypersensitivity reactions, a foreign allergen(s) is proposed to be recognized by IgE, which binds to its receptor Fc epsilon RI (FcεRI) on mast cells, thereby activating the mast cells and releasing highly active immune mediators [59] (Figure 1). The reactions often occur within minutes of the crosslinking of IgE to FcεRI receptors. Subsequently, the mediators secreted by mast cells can induce a late-phase reaction, usually 2–6 h after initiation, with a peak in activity after 6–9 h [60].

The second proposed mechanism, “the complement-receptor-mediated hypersensitivity,” can be initiated by the binding of allergens in vaccines and IgG or IgM and then activate the complement system to produce anaphylatoxins (e.g., C3a, C4a, and C5a) (Figure 1). These complement peptides can bind to complement receptors on mast cells, and then mast cell degranulation results in the release of immune mediators. In contrast to the IgE-dependent pathway, this proposed mechanism of hypersensitivity reaction does not involve IgE antibodies against allergens [61,62].

Third, several studies have suggested that the binding of allergens to MRGPRX2 (the mastocyte-related G-protein coupled receptor X2) protein, a class of G-protein-coupled receptors expressed on mast cells, may directly trigger mast cell activation and participate in non-IgE-mediated reactions [63] (Figure 1). It has been found that many molecules, such as antimicrobial host defense peptides, neuropeptides, and cationic amphiphilic drugs, could be the allergens for the induction [64].

However, no convincing evidence has demonstrated that MRGPRX2 or complements are involved in COVID-19-vaccine-induced immediate hypersensitivity reactions. An unknown mechanism may be involved in immediate hypersensitivity reactions induced by COVID-19 vaccines, which could trigger mast cell degranulation or other immune cell activations (Figure 1).

Many of the immediate hypersensitivity reactions are considered IgE-mediated, supported by skin prick tests and specific IgE levels [65]. Several excipients have been suggested to trigger the production of specific IgE antibodies and cause mast cell activation [65]. Patients with specific IgE antibodies against vaccine antigens may have higher risks of hypersensitivity reactions [66]. Several studies have attributed the development of hypersensitivity to increased specific IgE levels towards vaccine antigens. However, increased IgE levels may be a false-positive result in atopic individuals [48,67,68]. It is proposed that both IgE-mediated and non-IgE mediated pathways are involved in vaccine-induced immediate hypersensitivity reactions. Further studies are needed to investigate the mechanisms of COVID-19-vaccine-induced immediate hypersensitivity reactions.

## 6. Potential Allergens for COVID-19-Vaccine-Induced Immediate Hypersensitivity Reactions

### 6.1. Vaccine Excipients

COVID-19-vaccine-induced immediate hypersensitivity reactions have been proposed to be attributed to excipients. Polyethylene glycol (PEG), tromethamine, polysorbate 80, and ethylenediaminetetraacetic acid (EDTA) are excipients in the mRNA and viral vector COVID-19 vaccines [69]. The potential allergens that trigger hypersensitivity reactions following COVID-19 vaccinations are listed in Table 2.

PEGs, the well-known excipient of mRNA COVID-19 vaccines (including BNT162b2 and mRNA-1273) [70,71,72], are used in common products with limited knowledge of their potential to develop sensitization. Another excipient, tromethamine, used in the mRNA-1273 vaccine [71] is also found in cosmetics and some drugs [22]. A recent study estimated that 0.01% of allergic reactions to PEG and patients with skin disorders are highly significant for the development of PEG allergies. In patients with vaccine allergies, 0.09% of 1,055,677 doses were attributed to PEG [73]. Polysorbate 80, an excipient for COVID-19 vaccines (including the AZD1222 [AstraZeneca] and Ad26. COV2. S [Johnson & Johnson] vaccines), has been used in vaccines against hepatitis B and influenza and shows cross-reactivity with PEG because of its common structures [72].

Theoretically, specific IgE antibodies can recognize excipients and then present to the FcεRI receptors on mast cells or basophils [65]. The IgE-antigen complex triggers the degranulation of mast cells or basophils, resulting in the release of cytokines and other mediators, such as histamine and prostaglandins. Cytokines and other mediators can further promote CD4+Th2 responses and mast cell expansion and potentiate mast-cell-mediated prostaglandin actions. Together with immune mediators, they cause allergic symptoms that may eventually lead to anaphylaxis [22,74,75]. There are some studies suggesting that PEG can bind to specific antibodies and activate the complement system [76,77]. Owing to its high molecular weight, PEG bound by IgE cannot be phagocytosed but may trigger the complement pathway and promote opsonization [78].

In contrast, some studies have supported non-IgE-mediated immediate hypersensitivity reactions. A recent study reported that one patient with anaphylaxis related to the PEG of the BNT162b2 vaccine had negative skin test results and basophil activation tests, and no measurable specific IgE and IgG to PEG [79]. The study suggested that complement-mediated pathways or MRGPRX2-receptor-mediated pathways may be involved in immediate hypersensitivity [79]. The detailed mechanism needs to be further investigated. In addition, a case series study utilized a skin prick test and a basophil activation test to identify the main culprit excipient in patients with mRNA COVID-19-vaccine-related anaphylaxis. The results showed that only 1 of 11 patients who developed anaphylaxis was favorable to the mRNA vaccine via the skin prick test and had negative results for PEG and polysorbate. In contrast, all the patients showed a positive result on the basophil activation test for the mRNA COVID-19 vaccines, and 10 out of 11 were positive for PEG [80].

The main culprits have been initially suggested to be the excipients; however, almost all patients evaluated for immediate hypersensitivity reactions post-COVID-19-vaccination have shown negative allergy workups and tolerance to subsequent exposure(s). These observations indicate that nonallergic reactions or tolerance mechanisms for vaccine excipients might be found in patients.

### 6.2. Vaccine Antigens

Very limited evidence supports the role of the active product in the vaccine itself as an allergen. The spike glycoprotein is the most commonly used target for COVID-19 vaccines. It is composed of surface S1 and transmembrane S2 subunits. The surface S1 subunit is divided into five parts: the N-terminal domain, the C-terminal or receptor-binding domain, a receptor-binding motif, and two subdomains. On the other hand, the transmembrane S2 subunit is divided into an N-terminal hydrophobic fusion peptide, two repeats of heptads, a transmembrane domain, and a cytoplasmic tail [81]. Selvaraj et al. reported that the vaccine’s main component, spike glycoprotein, shares structural similarities with water-soluble glycoproteins that cause food allergies [82,83].

The vaccine antigen in the mRNA COVID-19 vaccines is designed for encoding the COVID-19 spike protein trimer RBD-binding subunit [84]. After vaccination, the mRNA vaccine antigen enters the cytoplasm of the host cells and attaches to ribosomes to generate COVID-19 viral spike proteins. The spike trimer subunit attracts CD4+ or CD8+ T cells to activate an immune response. CD4+ T cells can then release interferon-γ (IFN-γ) and interleukin-2 (IL-2), IL-4, and IL-5 to stimulate B cell differentiation and antibody production, whereas CD8+ T cells release cytotoxic molecules, such as IFN-γ, granzyme B (GZMB), and perforin, to amplify the immune response. Several studies have shown that mRNA COVID-19 vaccines (including BNT162b2 and mRNA-1273) predominantly induce CD4+Th1 cell proliferation [[3],[10],[85],[86],[87],].

Viral vector vaccines use genetically engineered viruses, such as AZD1222 (AstraZeneca) and Ad26. COV2. S (Johnson & Johnson), with inserts encoding COVID-19 spike protein. Upon entering the host cell, the viral vector uses host cell enzymes to produce proteins and induce immune responses [10,86]. The viral vectors are not pathogenic, yet either they or the translated viral proteins can still be recognized as foreign molecules and elicit immune responses, leading to neutralization and decreased vaccine efficacy if the following boost is the same formulation.

In addition, the epitopes of COVID-19 are similar to those of different aeroallergens [88]. Recently, several tools have been used to predict the allergenicity of some COVID-19 vaccine components. The potential allergens found were motifs and fusion peptides [82]. Some aeroallergens with structures similar to those of the components of COVID-19 vaccines may have previously caused sensitization and produced aeroallergen-specific IgE antibodies. Upon administration of the vaccine, antibodies may react to the vaccine components and then trigger mast cell and basophil degranulation [82,89]. Further experimental verification is required to investigate the mechanism in detail.

The detailed mechanism underlying inactivated COVID-19-vaccine-induced immediate hypersensitivity is still not fully understood. Inactivated COVID-19 vaccines are prepared following a conventional vaccine development approach. The whole virus is propagated in the medium and inactivated by various substances. Beta-propiolactone, a widely known viral activator, is commonly used. The inactivated virus is then adsorbed and sterilized prior to human use. These inactivated COVID-19 vaccines can be taken up by dendritic cells upon administration and produce an immune response. Subsequently, T-cell-dependent B cell differentiation produces antibodies against the antigen [90,91,92]. Few studies have been published on immediate hypersensitivity related to inactivated COVID-19 vaccines. One case series reported 12 patients who developed anaphylaxis after the first dose of the CoronaVac COVID-19 vaccines [93]. However, most of these patients were negative for both prick and intradermal tests and had normal tryptase levels [93]. The pathomechanism and main culprit compounds for inducing hypersensitivity by inactivated COVID-19 vaccines require further investigation.

## 7. Allergen Testing in COVID-19-Vaccine-Induced Immediate Hypersensitivity

Worldwide, specific laboratory tests for identifying and confirming the allergens in COVID-19 vaccines are unavailable. By comparison, in vivo allergen testing has been widely used to confirm the diagnosis of hypersensitivity reactions to food and drugs. The skin, basophil activation, and cysteinyl-leukotriene release tests are the most common methods used to identify culprit components in patients with immediate hypersensitivity reactions [94,95]. For example, the basophil activation test has shown a sensitivity of 55–80% and specificity of 80–96% for drug- and vaccine-induced immediate hypersensitivity reactions [96,97,98,99]. The skin prick test (SPT) and intradermal skin test (IDT) are reliable with acceptable sensitivity and specificity for hypersensitivity reactions caused by foods or drugs.

Table 3 lists the studies that investigated allergen tests for immediate hypersensitivity reactions to COVID-19 vaccines. These tests aim to help to identify the main causative allergic compound(s) [40,100,101,102,103,104,105,106]. However, the majority of these allergen test results were negative, suggesting that the mechanism of hypersensitivity reactions induced by COVID-19 vaccines may be more complicated than previously thought [100]. Further studies are needed to improve the accuracy and sensitivity of allergen testing, including increased detection of the possible compounds of COVID-19 vaccines (such as tromethamine, EDTA, or spike protein), dosage adjustment, or increased additional cell markers (e.g., MRGPRX2).

## 8. Risk Factors of COVID-19-Vaccine-Induced Immediate Hypersensitivity Reactions

Patients with a medical history of allergies were associated with further episodes of hypersensitivity reactions. After the administration of a reported 1893,360 first doses of the mRNA COVID-19 vaccine (BNT162b2) [108], the U.S. Center for Disease Control and Prevention (CDC) identified 21 cases of anaphylaxis [16]. Among them, 17 (81%) patients with anaphylaxis had a history of allergic reactions induced by drugs or medical products, foods, and insect stings, and 7 (33%) had previously experienced an episode of anaphylaxis induced by vaccines, including vaccinations against rabies and the influenza A (H1N1) vaccine. This study suggested that patients with a history of allergic reactions induced by drugs, foods, insect stings, or another vaccine may have a higher risk of mRNA COVID-19-vaccine-induced anaphylaxis [16].

In addition, Shavit et al. further reported that a higher rate of allergic reactions to the BNT162b2 vaccine was identified among patients with a history of sensitivity to aeroallergens, insect bites, food, latex, or drugs [109]. Recently, Chu et al. performed a systematic review and meta-analysis of case studies and found that the risk of immediate hypersensitivity reactions or anaphylaxis related to the second dose of an mRNA COVID-19 vaccine was lower among persons who experienced an immediate allergic reaction to their first dose (absolute risk, 0.16% [95% CI, 0.01–2.94%]) [33].

Genetic polymorphisms have also been proposed to be associated with immediate hypersensitivity reactions [110]. So far, there are still no published reports on genetic association with COVID-19-vaccine-induced immediate hypersensitivity reactions, but this may be a potential worthwhile research area.

## 9. Treatment and Prevention Strategies for COVID-19-Vaccine-Induced Immediate Hypersensitivity Reactions

### 9.1. Treatment Strategies

The management of COVID-19-vaccine-induced immediate hypersensitivity reactions is similar to other drug-induced immediate hypersensitivities and guided by three steps. The first step is to recognize the possibility of hypersensitivity to the vaccine based on medical history and clinical presentation. The second step is to rule out the possibility of systemic anaphylactic reactions to the vaccine excipients, such as PEG and polysorbate 80. The last step is medical management, which depends on the needs of patients.

Patients with COVID-19-vaccine-induced immediate hypersensitivity limited to cutaneous involvement, such as a mild skin rash, can be managed with antihistamines for the symptomatic relief of itching [18,111] Early administration of epinephrine is suggested for patients with vaccine-induced anaphylaxis (Figure 2). Corticosteroids are also helpful for the control of vaccine-induced immediate hypersensitivity reactions. Corticosteroids can downregulate the transcription of pro-inflammatory mediators in immune cells, such as mast cells, with the maximal theoretical effect observed 2 h after administration. Corticosteroids function by blocking glucocorticoid receptors to inhibit anaphylactic reactions. They further decrease mast-cell-mediated histamine release and promote anti-inflammatory mediator production [112]. However, systemic corticosteroids are not recommended because of the limited evidence of their efficacy for anaphylaxis [113].

Monoclonal antibodies against IgE, such as omalizumab, which blocks the binding of IgE to the FcεRI receptor, have potential use in patients with anaphylaxis based on a current clinical trial study [114]. Recently, IL-4 and IL-13 inhibitors such as dupilumab have been used to treat anaphylaxis by inhibiting mast cell expansion. These biological agents may be adjuncts in difficult-to-treat patients with immediate hypersensitivity and anaphylaxis [115,116]. Further clinical trials are needed to determine whether these medicines and biologic agents can be successfully used to treat patients with COVID-19-vaccine-induced immediate hypersensitivity or anaphylaxis.

### 9.2. Prevention Strategies

Currently, there are no methods to precisely prevent hypersensitivity reactions induced by COVID-19 vaccines, especially following the first dose. Based on clinical evaluation, patients with the following conditions may need to avoid using mRNA COVID-19 vaccines: (1) a history of anaphylaxis with injected drugs or vaccines containing PEG, (2) a history of anaphylaxis to oral or topical medications containing PEG, (3) a history of idiopathic recurrent anaphylaxis, (4) a confirmed or suspected mRNA COVID-19 vaccine allergy, and (5) a confirmed allergy to PEG or homologs [70,111,117]. In patients with hypersensitivity reactions or anaphylaxis following the first dose of an mRNA COVID-19 vaccine, it may be safe to administer fractionated doses (1/3 or 2/3) [70] or other formulations, such as protein-based vaccines. Patients at high risk for anaphylactic reactions have been suggested to refer to receive COVID-19 vaccination under medical supervision [109,118]. The recommendations help, through skin tests or in vitro tests, to identify the primary causative allergen of hypersensitivity reactions to these vaccines [70]. In addition, Smola et al. reported two cases of angioedema with or without urticarial rash after receiving the first dose of the mRNA-1273 COVID-19 vaccine. Tolerance to the second vaccination with mRNA-1273 COVID-19 after pretreatment with omalizumab was noted. This suggested that omalizumab can prevent hypersensitivity reactions induced by mRNA COVID-19 vaccines [119]. In practice, allergists must evaluate patients at high risk of hypersensitivity reactions induced by COVID-19 vaccines and consider the contraindications, risks, and benefits.

## 10. Conclusions

During the COVID-19 global pandemic, vaccination has significantly impacted controlling its spread. State-of-the-art technology has enabled the development of vaccines against certain diseases. Current published studies have consistently promoted the administration of third or fourth doses of the COVID-19 vaccines. This review article discussed the possible components, proposed mechanisms, prevention, and treatment strategies for immediate hypersensitivity reactions induced by COVID-19 vaccines. Investigations into allergens in the components of vaccines can help clinicians deal with the dilemmas of administering subsequent doses. Further studies are being conducted to elucidate the immunopathogenic mechanisms for preventing and treating COVID-19-vaccine-induced hypersensitivity reactions.

## Figures and Tables

**Figure 1 biomedicines-10-01260-f001:**
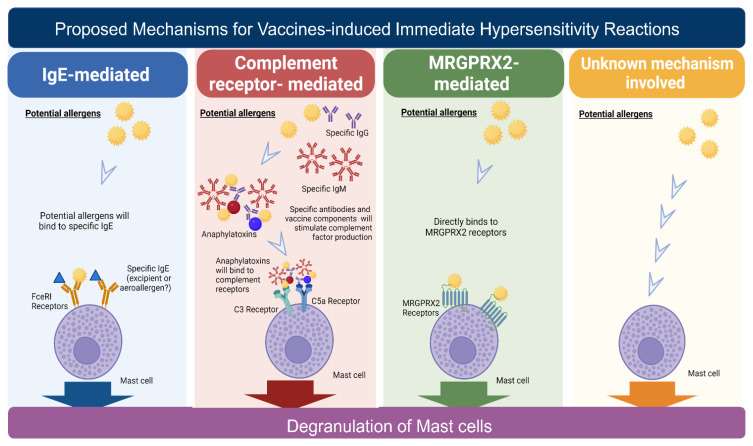
**Proposed mechanisms****for immediate hypersensitivity reactions.** There are four proposed mechanisms for immediate hypersensitivity: IgE-mediated, complement-receptor-mediated, Mas-related G protein-coupled receptor X2 (MRGPRX2)-mediated mast cell activation, and an unknown mechanism. Binding of allergens from the components of vaccines to antibodies or receptors may initiate the hypersensitivity reactions. The specific IgE antibodies recognize the active components or excipients of the vaccines. IgE antibodies are then coupled with receptor-FcεRI on the mast cells, resulting in mast cell degranulation. These specific IgE antibodies may be brought by previous exposure to allergens in cosmetics, drugs, aeroallergens, or food. Vaccine components may activate the complement-receptor-mediated pathway and induce anaphylatoxins that could be recognized by complement receptors on the mast cells. In addition, binding of the vaccine components and excipients to MRGPRX2 receptor may directly activate mast cells. Furthermore, immediate hypersensitivity reactions may be induced by an unknown mechanism. These proposed mechanisms could lead to mast cell degranulation and the release of effector mediators. Abbreviation: MRGPRX2, Mas-related G protein-coupled receptor X2; IgE, immunoglobulin E; PEG, polyethylene glycol.

**Figure 2 biomedicines-10-01260-f002:**
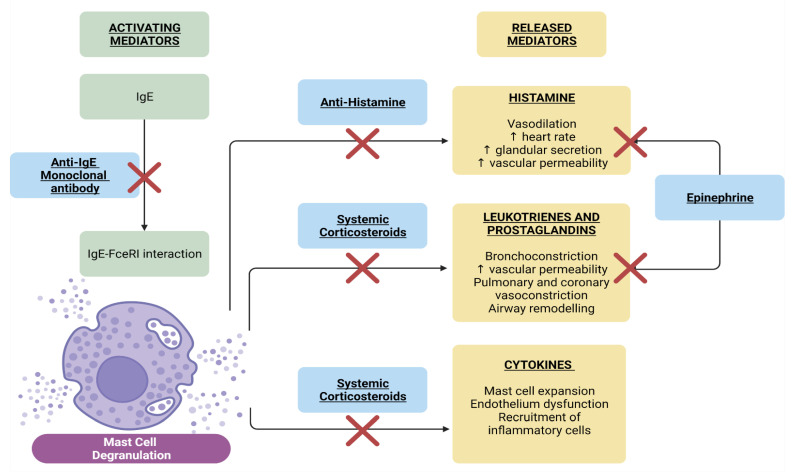
Proposed mechanism and therapeutic targets of immediate hypersensitivity reactions. For vaccine-induced immediate hypersensitivity reactions, potential targets can be used to attenuate mast cell degranulation through medicines or biologic agents. Anti-IgE monoclonal antibodies inhibit IgE–FcεRI interaction that initiates degranulation. Epinephrine acts as a physiologic antagonist of the effector mediators. Effector mediators released are inhibited by antihistamines and systemic corticosteroids. The symbol “X” represents it can block the pathway. ↑: Increase. Abbreviation: IgE, immunoglobulin E; FcεRI, Fc epsilon receptor.

**Table 1 biomedicines-10-01260-t001:** Incidence rates of anaphylactic and nonanaphylactic hypersensitivity reactions to COVID-19 vaccines.

Type of Reaction	Number of Participants	Number of Anaphylactic Reactions	Type of Vaccine	Incidence of Reactions (per One Million)	Reference
anaphylactic					
	890,604	15	mRNA-1273; BNT162b2	17	[35]
	4,041,396	10	mRNA-1273	37.1	[29]
	1,893,360	21	BNT162b2	11	[36]
	1116	1	BNT162b2; mRNA-1273	890	[37]
	283	5	mRNA-1273 and AZD1222	17,668	[38]
nonanaphylactic					
	277	14	BNT162b2	50,540	[39]
	5589	1391	AZD1222 (Astra Zeneca)	248,880	[39]
	5574	6	BNT162b2	1070	[40]
	3170	11	BNT162b2	3470	* [41]
	1,893,360	83	BNT162b2	43.8	* [36]
	877	10	BNT162b2	11,400	[42]
	1116	7	BNT162b2; mRNA-1273	6270	[37]
	74	35	BNT162b2	472,973	[23]

* Nonanaphylactic reactions were classified under skin rashes, including hives, pruritus, and eczematous papules.

**Table 2 biomedicines-10-01260-t002:** The potential allergenic components and excipients of COVID-19 vaccines.

Type of COVID-19 Vaccine	Vaccine Name (Manufacturer)	Potential Allergenic Components and Excipients	Function
mRNA vaccine	BNT162b2 (BioNTech- Pfizer)	2-[(polyethylene glycol[PEG])-2000]-N,Nditetradecylacetamide (ALC-0159)	Surfactant
mRNA vaccine	mRNA-1273 (Moderna)	SM-102, 1,2-dimyristoylrac-glycero-3-methoxypolyethyleneglycol-2000 [PEG2000-DMG] Tromethamine	Surfactant
mRNA vaccine	CvnCoV (CureVac)	PEGylated lipid	Surfactant
Viral vector vaccine	AZD1222 (Astra Zeneca)	Polysorbate 80 EDTA	Surfactant
Viral vector vaccine	Ad26.COV2.S (Johnson and Johnson)	Polysorbate 80	Surfactant
Viral vector vaccine	Gam-COVID-Vac (Sputnik V)	Polysorbate 80 EDTA	Surfactant
Protein-based vaccine	NVX-CoV2373 (Novavax)	Polysorbate 80	Surfactant
Protein-based vaccine	Sanofi/GSK (Sanofi Pasteur and GSK)	Polysorbate 20	Surfactant
Inactivated vaccine	CoronaVac (Sinovac)	Not available	Not available

**Table 3 biomedicines-10-01260-t003:** Allergen testing in immediate hypersensitivity reactions by COVID-19 vaccines.

Method	Number of Participants	Number of Positive Results	Reference
SPT and IDT	6 patients; 18 controls	18 (BNT162b2)	[40]
SPT	4 patients	0	[100]
SPT and IDT	2 patients	2 (mRNA-1273)	[101]
SPT	131 patients	2 (PEG3350)	[102]
SPT	1 patient	0	[103]
SPT	15 patients	1 (PEG3350)	[107]
BAT	1 patient	1 (PEG)	[104]
SPT	1 patient	1 (PEG4000)	[105]
SPT and IDT	60 patients; 55 controls	4 (BNT162b2) 1 (PEG2000)	[106]

Abbreviation: BAT, Basophil activation test; SPT, skin prick test; IDT, intradermal skin test.

## Data Availability

Not applicable.

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
