# Peer review of "Immediate Hypersensitivity Reactions Induced by COVID-19 Vaccines: Current Trends, Potential Mechanisms and Prevention Strategies"

_biomedicines, 2022, doi:10.3390/biomedicines10061260_

Round 1

Reviewer 1 Report

in the current manuscript "Immediate hypersensitivity reactions induced by COVID-19 2 vaccines: current trends, potential mechanisms and prevention 3 strategies", C. Preclaro et. al describes the possible mechanisms of hypersensitivity events that happen upon available COVID-19 vaccination and the possibility of the correlation between excipients and this events. 

the article is well written with good language and need minor revision,

  1. table 1, the references should be at the end column not the first.
  2. the two parts of Table 1 should be combined together and have a column named (Type of reaction), then add in each row ( anaphylactic or non-anaphylactic).
  3. table 1 and 3, no need to write the reference twice, just the numbered reference is fine 
  4.  

Reviewer 2 Report

Preclaro et al. provide an overview on immediate hypersensitivity reaction (iHR) induced by COVID-19 vaccines and discuss the potential mechanisms, diagnostic tests and prevention strategy. The article is well-written and aims to summarize the prevailing literature. My main concern is that, concerning the mechanism, there is very limited evidence (if any) to support the different hypothesis that are put forward. Therefore it remains important to clearly discriminate hypothesis from evidence. I would urge the authors to critically revise the statements and to make sure that only those based on firm evidence are included as such, whilst others clearly read as being (unproven) hypothesis.

A section I missed, is that on the differential diagnosis of immediate hypersensitivity reactions. Many are not confirmed (for instance using Brighton collaboration criteria) or post-factum attributed to alternative diagnosis such as vasovagal syncope, vocal cord dysfunction, exacerbation of existing chronic spontaneous urticaria, anxiety, etc.

Almost all reported incidences come from passive reporting systems (such as VAERS) which may underestimate the true burden. Limited prospective studies have been performed and these indicate a much higher rate of iHR (Blumenthal JAMA et al. 2021), possibly due to a nocebo effect (Amanzio et al. Lancet Reg Health Eur 2021 and Ieven et al. Vaccines 2022).

Another recent study that merits inclusion is Chu et al. JAMA int med 2022.

Abstract. The authors state in the abstract that vaccine-induced HRs are the main reason for vaccine hesitance. However, this is not supported by data and reads as a justification of the work. I would suggest to rephrase this to a more neutral statement (or provide in the manuscript the evidence supporting this statement).

Similar, thus far there has been no convincing evidence that MRGPRX2, or complement to be involved in COVID19 (or any) vaccine-induced iHR. This has been put forward as an explanation for non-IgE-mediated anaphylaxis, but none have been supported by clinical, biochemical or in vitro evidence.

Next, the main culprits have been initially suggested to be the excipients, but almost all patients evaluated for an iHR post-COVID-19 vaccination demonstrate to have negative allergy workups and tolerate subsequent exposure(s), indicating the non-allergic nature of the majority  (almost all) of the reactions.

The suggestion of a higher rate of anaphylaxis due to BNT162b2 versus Moderna should be substantiated as more likely different study setup could explain this (as suggested by the authors of these articles).

Preferably use of mRNA-1273 for Moderna if BNT162b2 is used for Pfizer-BioNTech vaccines.

Please add the limited available information on anaphylaxis after live-attenuated vaccines and protein-based vaccines (Nuvaxovid/Novavax).

In line 130, it is indicated that iHR a) occur rapidly and b) are also called Type I HR. For a), this seems logic as they are immediate. For b) this is incorrect as only IgE-mediated reactions should be termed type I HR.

Perhaps also add which classification is used (G&C)

Next, this section is a bit confusion due to the mentioning of mast cell implication in type II and III reactions, which has not been formally proven.

Finally, the paragraph ends with 3 possible explanations for iHR induced by vaccines. Please rephrase and reorder this paragraph.

In line 145-152, the authors describe complement receptors-mediated mast cell degranulation. The section reads as if there is formal proof for this mechanism with demonstration of involvement of C1q etc. Whilst this has been postulated many times, there is no actual evidence for this, especially in vaccine-induced iHR. It remains important to phrase this as a ‘proposed mechanism’, not only in the beginning of the paragraph but also throughout the text.

In figure 1, three mechanisms are proposed. But perhaps additional are present. It would be wise to add a fourth ‘unknown mechanism’ to indicate the gap of knowledge on this topic.

Spike protein is suggested in Fig 1 as a potential allergen in the IgE-mediated iHR. This would only hold true for live-attenuated vaccines or the protein-based vaccines. Next, there is no evidence for this. In the legend also mRNA, viral vector DNA is suggested. There is no evidence for this. Please modify so this is clear.

References 54, 39 and 55 are indicated to support vaccine components can induce IgE and trigger type I iHR upon re-exposure. Perhaps the authors could rephrase this to vaccine excipients (as reviewed in Caballero et al JACI Pract 2021) whilst very limited evidence supports the role of the active product in the  vaccine itself as an allergen.

Line 202; this sentence seems incorrectly formulated.

Line 215-217: please add a reference after this statement

Line 230-232: these in vitro findings do not prove that PEG is one of the allergic excipients. These data have not been reproduced nor explained and limited/no negative controls were evaluated (or reported). Perhaps rephrase this statement to “the authors concluded that …“ … “however …”.

Line 268-271: Perhaps rephrase to “… antibodies have been proposed to react to spike protein …” . As it is written now, it seems proven, which is incorrect. 20% of the population is sensitized for aeroallergens, yet limited reactions occur and no firm correlation (and especially no causality) has been inferred.

The section on PEG or polysorbate 80 is limited and could be improved.

The section on risk factors should be improved. Currently only the initial CDC report (ref 14) is mentioned, whilst much more data has been gathered on this topic. The section on genetic predisposition is speculation and should/could be omitted instead (or limited to one sentence).

In the treatment/prevention section the second step (“to rule out the possibility of systemic reaction to the vaccine”) is unclear. Do the authors mean to rule out an underlying allergy.

In this section also acute treatment (antiH1, epinephrine) is mixed up with prevention (omalizumab) and should be more distinct. Moreover, the statement that “Early administration of epinephrine to reduce the risk of recurrence … remains the gold standard” is incorrect.

The authors provide substantial information on corticosteroids whilst their role in the acute treatment of anaphylaxis is debated and currently put as a later line treatment (EAACI) or (almost always) omitted (UK resuscitation guidelines). Moreover, evidence from patients receiving chronic steroid treatment indicates blunting of vaccine-induced immune response making the addition of these statement rather contra-productive.

What do the authors mean by “based on allergen evaluation” line 370 ?

Best evidence for tolerance of vaccination in these ‘at-risk’ population comes from Shavit et al. JAMA Netw Open 2021 and Ieven et al Vaccines 2022, The ENDA/EAACI provides a position and guidance, albeit with limited data to support this.
